# Ensemble Learning for Disease Prediction: A Review

**DOI:** 10.3390/healthcare11121808

**Published:** 2023-06-20

**Authors:** Palak Mahajan, Shahadat Uddin, Farshid Hajati, Mohammad Ali Moni

**Affiliations:** 1College of Engineering and Science, Victoria University, Sydney, NSW 2000, Australia; 2School of Project Management, Faculty of Engineering, The University of Sydney, Forest Lodge, NSW 2037, Australia; 3School of Health and Rehabilitation Sciences, Faculty of Health and Behavioural Sciences, The University of Queensland, St. Lucia, QLD 4072, Australia

**Keywords:** machine learning, bagging, boosting, stacking, voting, disease prediction

## Abstract

Machine learning models are used to create and enhance various disease prediction frameworks. Ensemble learning is a machine learning technique that combines multiple classifiers to improve performance by making more accurate predictions than a single classifier. Although numerous studies have employed ensemble approaches for disease prediction, there is a lack of thorough assessment of commonly used ensemble approaches against highly researched diseases. Consequently, this study aims to identify significant trends in the performance accuracies of ensemble techniques (i.e., bagging, boosting, stacking, and voting) against five hugely researched diseases (i.e., diabetes, skin disease, kidney disease, liver disease, and heart conditions). Using a well-defined search strategy, we first identified 45 articles from the current literature that applied two or more of the four ensemble approaches to any of these five diseases and were published in 2016–2023. Although stacking has been used the fewest number of times (23) compared with bagging (41) and boosting (37), it showed the most accurate performance the most times (19 out of 23). The voting approach is the second-best ensemble approach, as revealed in this review. Stacking always revealed the most accurate performance in the reviewed articles for skin disease and diabetes. Bagging demonstrated the best performance for kidney disease (five out of six times) and boosting for liver and diabetes (four out of six times). The results show that stacking has demonstrated greater accuracy in disease prediction than the other three candidate algorithms. Our study also demonstrates variability in the perceived performance of different ensemble approaches against frequently used disease datasets. The findings of this work will assist researchers in better understanding current trends and hotspots in disease prediction models that employ ensemble learning, as well as in determining a more suitable ensemble model for predictive disease analytics. This article also discusses variability in the perceived performance of different ensemble approaches against frequently used disease datasets.

## 1. Introduction

Ensemble learning is a machine learning approach that attempts to improve predictive performance by mixing predictions from many models. Employing ensemble models aims to reduce prediction generalisation error [1]. The ensemble technique decreases model prediction error when the base models are diverse and independent. The technique turns to the collective output of individuals to develop a forecast. Despite numerous base models, the ensemble model operates and performs as a single model [2]. Most real data mining solutions employ ensemble modelling methodologies. Ensemble approaches combine different machine learning algorithms to create more accurate predictions than those made by a single classifier [3]. The ensemble model’s main purpose is to combine numerous weak learners to form a powerful learner, boosting the model’s accuracy [4]. The main sources of the mismatch between real and predicted values when estimating the target variable using any machine-learning approach are noise, variation, and bias [5].

Disease diagnosis refers to the process of determining which disease best reflects a person’s symptoms. The most challenging issue is diagnosis because certain symptoms and indications are vague, and disease identification is vital in treating any sickness [6]. Machine learning is a field that can help anticipate disease diagnosis based on prior training data [7]. Many scientists have created various machine learning algorithms to effectively identify a wide range of conditions. Machine learning algorithms can create a model that predicts diseases and their treatments [7]. Because of the vast amount of data available, disease prediction has become a significant research subject. Using these databases, researchers create disease prediction models for decision-making systems, allowing for better disease prediction and treatment at an early stage. Early diagnosis and timely treatment are the most effective ways to lower disease-related mortality rates [8]. As a result, most medical scientists are drawn to emerging machine learning-based predictive model technologies for disease prediction.

Diabetes, skin cancer, kidney disease, liver disease, and heart conditions are common diseases that can significantly impact patients’ health. This research explores the literature for disease prediction models based on these diseases. We initially identified several types of disease prediction models by reviewing the current literature based on the five disease categories considered using a search strategy. The scope of this study is to find essential trends among ensemble approaches used on various base model learners, their accuracy, and the diseases being studied in the literature. Given the increasing relevance and efficiency of the ensemble approach for predictive disease modelling, the field of study appears to be expanding. We found limited research that thoroughly evaluates published studies applying ensemble learning for disease prediction. As a result, this study aims to uncover critical trends across various ensemble techniques (i.e., bagging, boosting, stacking, and voting), their performance accuracies, and the diseases being researched. Furthermore, the benefits and drawbacks of various ensemble techniques are summarised. The outcomes of this study will help researchers better understand current trends and hotspots in disease prediction models that use ensemble learning and help them establish research priorities accordingly.

Following is a summary of the remaining sections of the article: the ensemble methods (bagging, boosting, stacking, and voting ensemble approaches) are briefly outlined in the Section 2. The Section 3 of the document contains information on the scientific material that was analysed from 2016 to 2023. A summary of the articles that used at least two of the four classical ensemble approaches for one of the five major chronic diseases considered in this study, and their performance accuracy, benefits, and drawbacks are then outlined in the Section 4. We also provided a comparison of the usage frequency and top performances of the ensemble classes based on diseases taken into consideration in this section. Then, we discuss the findings of this study, followed by the Section 6.

## 2. Ensemble Learning

Ensemble learning is a machine learning approach that combines predictions from multiple models to increase predictive performance [7]. Ensemble techniques integrate various machine learning algorithms to make more accurate predictions than a single classifier. The use of ensemble models is intended to reduce the generalisation error. This technique reduces model prediction error when the base models are diverse and independent. As outlined in Figure 1, The approach relies on the collective output of individuals for generating forecasting. Although several base models exist, the ensemble model works and behaves as a single model [7].

The main sources of the mismatch between actual and predicted values are noise, variation, and bias when estimating the target variable using any machine learning approach. As a result, ensemble techniques combine several machine learning algorithms to create more accurate predictions than those made by a single classifier [1]. The ensemble model combines multiple models to reduce the model error and maintain the model’s generalisation. Bagging, boosting, stacking, and voting are the four major classes of ensemble learning algorithms, and it is critical to understand each one and incorporate them into every predictive modelling endeavour [9].

### 2.1. Bagging

Bagging is aggregating the predictions of many decision trees that have been fit to different samples of the same dataset. Ensemble bagging is created by assembling a series of classifiers that repeatedly run a given algorithm on distinct versions of the training dataset [10]. Bagging, also known as bootstrapping [7], is the process of resampling data from the training set with the same cardinality as the starting set to reduce the classifier’s variance and overfitting [11]. Compared to a single model, the final model should be less overfitted. A model with a high variance indicates that the outcome is sensitive to the practice data provided [12]. As a result, even with more training data, the model may still perform poorly and may not even lower the variance of our model. The overall framework for bagging is presented in Figure 2.

Bootstrapping is like random replacement sampling that can provide a better understanding of a data set’s bias and variation [12]. A small portion of the dataset is sampled randomly as part of the bootstrap procedure. Random Forest and Random Subspace are upgraded versions of decision trees that use the bagging approach to improve the predictions of the decision tree’s base classifier [13]. For generating multiple split trees, this technique uses a subset of training samples as well as a subset of characteristics. Multiple decision trees are created to fit each training subset. The dataset’s distribution of attributes and samples is normally performed at random.

Another bagging technique is the extra trees, in which many decision trees combine forecasts. It mixes a vast number of decision trees, like the random forest. On the other hand, the other trees employ the entire sample while selecting splits at random. Although assembling may increase computational complexity, bagging can be parallelisable. This can significantly reduce training time, subject to the availability of hardware for running parallel models [5]. Because deep learning models take a long time to train, optimising several deep models on various training bags is not an option.

### 2.2. Boosting

Boosting algorithms use weighted averages to transform poor learners into strong ones. During boosting, the original dataset is partitioned into several subgroups. The subset is used to train the classifier, which results in a sequence of models with modest performance [7]. The elements that were incorrectly categorised by the prior model are used to build new subsets. The ensembling procedure then improves its performance by integrating the weak models using a cost function. It explained that, unlike bagging, each model functions independently before aggregating the inputs, with no model selection at the end. Boosting is a method of consecutively placing multiple weak pupils in a flexible manner. Intuitively, the new model focuses on the discoveries that have been shown to be the most difficult to match up until now, resulting in a good learner with less bias at the end of the process [10]. Boosting can be used to solve regression and identification problems, such as bagging. Figure 3 illustrates the framework of the boosting approach.

When compared to a single weak learner, strategies such as majority voting in classification problems or a linear combination of weak learners in regression problems result in superior prediction [14]. The boosting approach trains a weak learner, computes its predictions, selects training samples that were mistakenly categorised, and then trains a subsequent weak learner with an updated training set that includes the incorrectly classified instances from the previous training session [15]. Boosting algorithms such as AdaBoost and Gradient Boosting have been applied in various sectors [16]. AdaBoost employs a greedy strategy to minimise a convex surrogate function upper limit by misclassification loss by augmenting the existing model with the suitably weighted predictor at each iteration [10]. AdaBoost optimises the exponential loss function, whereas Gradient Boost extends this approach to any differential loss function.

### 2.3. Stacking

Stacking is an assembly method in which one or more base-level classifiers are stacked with a metalearner classifier. The original data is used as input to numerous distinct models in stacking [5]. The metaclassifier is then utilised to estimate the input as well as the output of each model, as well as the weights [7]. The best-performing models are chosen, while the rest are rejected. Stacking employs a metaclassifier to merge multiple base classifiers trained using different learning methods on a single dataset. The model predictions are mixed with the inputs from each successive layer to generate a new set of predictions [17]. Ensemble stacking is also known as mixing because all data is mixed to produce a forecast or categorisation. Multilinear response (MLR) and probability distribution (PD) stacking are the most advanced techniques. Groupings of numerous base-level classifiers (with weakly connected predictions) are widely known to work well. Nahar et al. [3] propose a stacking technique that employs correspondence analysis to find correlations between base-level classifier predictions. Figure 4 depicts the framework for the stacking approach.

The dataset is randomly divided into *n* equal sections in this procedure. One set is utilised for testing and the rest for training in the nth-fold cross-validation [8]. We derive the predictions of various learning models using these training testing pair subsets, which are then used as metadata to build the metamodel [18]. The metamodel produces the final forecast, commonly known as the winner-takes-all technique [5]. Stacking is an integrated approach that uses the metalearning model to integrate the output of base models. If the final decision element is a linear model, the stacking is also known as “model blending” or just “blending” [8]. Stacking involves fitting multiple different types of models to the same data and then using a different model to determine how to combine the results most efficiently [3].

### 2.4. Voting

The Voting Classifier ensemble approach is a strategy that aggregates predictions from numerous independent models (base estimators) to make a final prediction [19]. It uses the “wisdom of the crowd” notion to create more accurate predictions by taking into account the aggregate judgement of numerous models rather than depending on a single model. In the Voting Classifier, there are two types of voting: hard voting, in which each model makes a prediction, and soft voting, in which each model forecasts the probability or confidence ratings for each class or label. The final prediction is made by summing the expected probabilities across all models and choosing the class with the highest average probability [20]. Weighted voting allows multiple models to have different influences on the final forecast, which can be assigned manually or learned automatically based on the performance of the individual models. Because of this diversity, different models can affect the final prediction differently [19].

By combining the strengths of several models, the Voting Classifier increases overall performance and robustness, especially when distinct models have diverse properties and generate independent predictions. The Voting Classifier can overcome biases or limits in a single model and produce more accurate and trustworthy predictions by using the collective decision-making of numerous models. Overall, Voting Classifier is a versatile ensemble strategy that can be applied to a variety of machine learning applications by using the capabilities of different models to make more accurate predictions [7]. The Voting Classifier is a versatile ensemble approach in machine learning that provides a number of benefits. By integrating various models with diverse strengths and weaknesses, it enhances accuracy, robustness, and model diversity while minimising bias and variance in predictions.

The Voting Classifier’s ensemble nature improves model stability by decreasing overfitting and increasing model variety [21]. It offers various voting procedures like as hard voting, soft voting, and weighted voting, allowing for customisation based on the tasks and characteristics of specific models. Furthermore, the Voting Classifier can improve interpretability by analysing the contributions of many models, assisting in understanding the underlying patterns and decision-making process [22]. Overall, the Voting Classifier is an effective tool for enhancing predictive performance in various machine learning tasks.

## 3. Methods

This study examined the scientific literature for five major chronic diseases and the algorithms used for their predictive analytics. This study examined research articles from IEEE, ScienceDirect, PubMed, Springer, and Scopus from 2016 to 2023. The research studies reviewed here were based on five important disease prediction categories: heart, skin, liver, kidney, and diabetes, all of which are known to cause a variety of ailments. These are the major chronic diseases (MCDs) around the world. Governments worldwide are concerned about the disease burden caused by MCDs. Almost 1 in 2 Australians (47%) had one or more MCDs in 2017–2018, and nearly 9 in 10 (89%) of deaths in Australia in 2018 were associated with them [23]. The World Health Organization (WHO) considers MCDs an “invisible epidemic” that continues to be widely ignored and hinders many countries’ economic development [24]. For these reasons, we considered these diseases. Furthermore, significant research was conducted to identify publications that used two or more of the four classical ensemble approaches (bagging, boosting, stacking, and voting) for disease prediction.

Springer is the world’s largest online repository of scientific, technological, and medical periodicals. Elsevier’s Scopus is an online bibliometric database. It was picked for its high level of accuracy and consistency. PubMed supports the open discovery and retrieval of biomedical and life sciences publications to enhance individual and global health. More than 35 million links and summaries from biomedical publications are available in the PubMed database [25]. Nearly four million verified journal and conference articles are available in the IEEE Xplore digital collection to help you feed creativity, build on earlier research, and inspire new ideas. We followed a thorough literature search approach to discover all relevant publications in this study. On 30 May 2023, we used the following search phrases for this search.

“Disease prediction” and “ensemble method”“Disease prediction” and “ensemble machine learning”“Disease diagnosis” and “ensemble machine learning”“Disease diagnosis” and “ensemble learning”

The terms “disease prediction” and “disease diagnosis” were used with the “ensemble method” and “ensemble machine learning” in the sources, yielding a total of 1046 articles that included these keywords in the titles, abstracts, or author-selected keywords used in their articles. After deleting duplicates and considering English-written articles, we had 766 publications. We selected only those that used at least two of the four ensemble approaches of bagging, boosting, stacking, or voting for their predictive analysis and published in 2016–2023. This further screening reduced the number of selected articles from 766 to 45. Although there were publications on other diseases, only five disease categories were chosen for this study. Hence, the research articles picked were for these five disorders: heart, skin, liver, kidney, and diabetes. Figure 5 depicts the data-gathering technique and the number of articles chosen for various diseases.

## 4. Results

This study reviewed several articles from the current literature that predicted different major diseases. Ensemble techniques help improve the prediction accuracy of weak classifiers and perform better in predicting heart disease risk [4]. However, majority voting generates the most significant improvement in accuracy [4,7]. Another study by Ashri et al. [20] forecasted the risk of heart disease with an accuracy of 98.18%, whereas majority voting produced the best outcomes regarding increased prediction accuracy (98.38%) [27]. They used several classifiers to improve the performance, but ensemble approaches predicted the risk of heart disease more accurately than individual classifiers. Another study by Karadeniz et al. [28] developed and tested innovative ensemble approaches for predicting cardiac disease. They derived a classifier from random analysis of distance sequences and then used it as the base estimation of a bagging strategy to improve performance.

Chronic kidney disease (CKD) is a condition that destroys both kidneys, preventing them from completing their tasks [29]. Unfiltered wastes or poisons in the blood can cause high blood pressure, diabetes, and other disorders, finally paralysing the entire system and culminating in death [13]. A study by Basar and Akan [11] for CKD prediction used two ensemble approaches on four base learners to increase the underlying models’ classification performance. They are k-nearest neighbours, Naive Bayes and Decision Tree. This study found the empirical results of 100% prediction accuracy using the ensemble random subspace approach on the CKD dataset obtained from the UCI machine learning repository [13,30]. In their study, Eroğlu and Palabaş [30] also received a prediction accuracy of 99.15% for CKD, making the model more suitable and appropriate for accurately detecting CKD.

Verma et al. [16] presented a new technique that employs six different data mining classification approaches and then developed an ensemble approach that uses bagging, AdaBoost, and Gradient Boosting classifiers to forecast skin disease [16]. They then used four different ensemble approaches (bagging, boosting, stacking, and voting) for diverse base learners to improve the performance for skin diseases, resulting in an accuracy of 99.67% [31]. Pal and Roy [18] used different base learners for implementing ensemble approaches and found that the stacking model gave an accuracy of 100%. The base learners they used are Naïve Bayes (NB), k-nearest neighbours (KNN), support vector machine (SVM), artificial neural network (ANN), and random forest (RF).

The liver is the largest internal organ in the human body, weighing an average of 1.6 kg (3.5 pounds) [3]. Liver disease can be inherited (genetically) or caused by various circumstances, such as viruses and alcohol usage [32]. Singh et al. [33] applied machine learning techniques, including logistic regression (LR), extreme gradient boosting (XG-Boost), decision tree (DT), CatBoost, AdaBoost, and Light gradient boosting machine (LGBM) on selected features from the dataset for predicting liver disease. They found an accuracy of 88.55%, 86.75%, and 84.34% for RF, XGBoost, and LGBM, respectively.

Diabetes harms several bodily organs, leading to issues including eye issues, chronic renal disease, nerve damage, heart issues, and foot issues. Abdollahi and Nouri-Moghaddam [34] forecasted diabetes using a stacked generalisation technique that achieved an accuracy of 98.8% in diagnosing diseases. The Pima Indians Diabetes dataset was used in a diabetes study led by Kumari et al. [6]. They found great accuracy, precision, recall, and F1 score values of 79.04%, 73.48%, 71.45%, and 80.6%, respectively. Their suggested best classification model used an ensemble of four machine learning algorithms (RF, LR, and NB). These studies increase prediction accuracy by integrating higher-level and lower-level models. The stacking technique with metalearner sequential minimal optimisation (SMO) is the most accurate classifier, as revealed in a diabetes study by Singh and Singh [17], which achieved 79% accuracy and 78.8% precision.

Table 1, Table 2, Table 3, Table 4 and Table 5 summarise all reviewed articles considered in this study. The tables detail the names of the diseases, associated references and the ensemble machine-learning algorithms used for disease prediction. This study comprised 45 publications, and the optimal approach for each disease type is mentioned in the tables below. The primary focus will be comparing ensemble strategies, such as bagging, boosting, stacking, and voting and selecting the best clinically applicable framework to increase prediction accuracy of complex disease conditions.

### 4.1. Advantages and Limitations

Table 6 below details the benefits and drawbacks of this study’s four main classes of ensemble approaches.

### 4.2. Frequency and Accuracy Comparison

Table 7 compares usage frequency and best performance for bagging, boosting, stacking and voting algorithms used in the reviewed articles, as outlined in Table 1, Table 2, Table 3, Table 4 and Table 5. Although boosting has been frequently employed (37 of 45), the frequency of the most accurate in percentage is only 40.5%. Stacking, on the other hand, was used in 23 of the 45 studies, but the accuracy rate is 82.6%. Random forest and random subspace are the upgraded version of bagging, so in this comparison analysis, these approaches have been included in the bagging section. Bagging has produced the poorest outcomes, with the highest accuracy of only 26.8% with a usage frequency of 41 out of 45. Voting has also shown a good frequency of the best performance accuracy of 71.4%, but it appeared in seven out of 45 reviewed articles.

Table 8 compares the precision of bagging, boosting, stacking, and voting for the various MCDs considered in this research. Regarding heart disease, bagging showed the best accuracy in only 13.4% of cases. While stacking has a lower study count, it revealed the best performance at the highest times (80%). The papers considered for this article show that no kidney disease study has used stacking and voting algorithms, whereas, between bagging and boosting, bagging showed 83.4% times accuracy. For skin cancer and diabetes, the stacking algorithm revealed the best performance when it was used for prediction (100%). Voting has been used in only heart and diabetes studies with an accuracy of 75% and 66.7%, respectively. Additionally, overall stacking, used in only approximately half of the trials (23 times), revealed the best performance in percentage value (82.6%). Voting was used in seven studies only but has given a total best accuracy of 71.4% of cases.

Stacking is a popular and efficient strategy for heart disease, skin cancer, liver disease, and diabetes prediction. It can capture and combine varied predictions from base models, uncovering hidden patterns and complicated connections between risk factors and symptoms [10]. It also captures nonlinear relationships, taking advantage of the flexibility of various modelling techniques to capture nonlinear patterns more successfully than other ensemble methods, such as bagging or boosting [34]. Stacking can combine predictions from several base models trained on distinct subsets or representations of the data, exploiting the strengths of each model while limiting the impact of data heterogeneity. It can address class imbalance issues in imbalanced datasets by optimising performance on minority classes and incorporating models with various uncertainty handling strategies, such as imputation approaches or robust estimation methods [4,8,37]. Regardless of these advantages, like other machine learning methods, the performance of the stacking approach can be affected by various factors, including dataset attributes, clinical features, and modelling choices.

Ensemble approaches that balance performance and interpretability differently include bagging, boosting, stacking, and voting. Bagging combines many models, producing great accuracy but no comprehensive understanding of feature relationships. Boosting enhances the model by giving samples that were incorrectly classified as more weight, but it might make the model more complex and challenging to understand [32]. Stacking is more difficult to understand and uses a metalearner to combine many models. Forecasts are gathered through voting using the majority or weighted voting method, balancing accuracy, and interpretability [21,45]. The selection is based on the application’s requirements, considering accuracy, interpretability, and issue domain complexity.

### 4.3. Ensemble Performance against Datasets

Our reviewed 45 articles used 23 distinct datasets from various open-access sources, including Kaggle and UCI, as evident in Table 1, Table 2, Table 3, Table 4 and Table 5. The five most used datasets are UCI Cleveland Heart Disease [58], UCI Chronic Kidney [59], UCI Dermatology [60], UCI Indian Liver Patient [61], and Pima Indians Diabetes [62]. All four ensemble approaches considered in this study have been applied to these datasets. However, not each of these four ensemble approaches has been found as the best-performing against each dataset, as revealed in Figure 6. The voting approach was found as the best performing only for two datasets (Pima Indians Diabetes and UCI Cleveland Heart Disease). It does not show the highest accuracy for the other three datasets. Stacking has been found to offer the best accuracy in four datasets, except for the UCI Chronic Kidney. Only the boosting approach delivered the best accuracy for all five datasets. Bagging has been found as the best performing for three datasets.

## 5. Discussion

Through machine learning, numerous methods for the early identification of diseases are being produced and enhanced. The ensemble method in machine learning increases disease forecast accuracy by reducing bias and variation. As a result, it is the best way to identify diseases. Although there are other ensemble approach variations, we have considered bagging, boosting, stacking, and voting for disease prediction. Heart disease prediction has received more research attention than skin cancer, liver, diabetes, and kidney diseases, according to our study of the five chronic diseases. Our review also shows that the accuracy of the basic classifier can be improved by applying an ensemble method to other machine learning algorithms. Disease prediction models may assist doctors in identifying high-risk individuals, improving individual health outcomes and care cost-effectiveness.

Several strategies were used in the preprocessing stage to prepare the data for analysis. These techniques included test–train splitting, random oversampling to address the class imbalance, feature selection, k-fold validation, missing value handling, outlier elimination, scaling, data imputation, label encoding, resampling methods like SMOTE, feature engineering, feature extraction, discretisation, and PCA (Principal Component Analysis). Optimisation strategies and hybrid feature selection methods were also used in some approaches. Researchers use these strategies singly or in combination, depending on the specific data and analysis requirements. These preprocessing procedures were designed to ensure data quality, resolve missing values, deal with class imbalance, reduce dimensionality, and prepare the dataset for later analysis and modelling activities.

The stacking method is the best choice for selecting an ensemble approach that can produce more accurate results because it has demonstrated better forecast accuracy in the reviewed studies. This literature review showed that stacking had the highest performance accuracy, with 82.6% of the cases when researchers used this approach. The stacking approach predicted liver and diabetes diseases with 100% success, as revealed in this review study. Researchers used bagging (41 out of 45) and boosting (37 out of 45) more frequently than stacking (23 out of 45) and voting (7 out of 45), but stacking and voting showed better performance than the other two. In only 26.8% of the cases when it was used, bagging showed the best performance. For the boosting approach, it was only 40.5%. According to this literature review, the most popular strategy is stacking, which has produced more reliable outcomes than bagging and boosting despite being used in fewer studies. Voting is the second most popular strategy, producing better results than bagging and boosting but used less frequently compared with them.

Data leakage could lead to overly optimistic performance estimates, which could be a concern in ensemble techniques for disease identification. To address this issue, researchers used techniques such as suitable cross-validation [63], train–validation–test split [64], feature engineering precautions, and model stacking precautions [65]. A three-way data split divides the dataset into training, validation, and test sets, and cross-validation techniques such as k-fold ensure an accurate separation of training and testing data. Commonly used feature engineering procedures are scaling, imputation and feature selection which are performed during the cross-validation cycle to prevent leakage. Because of model stacking safety precautions, metalearners are educated on predictions that were not used during training. These methods aided in lowering the risk of data leakage to ensure the legitimacy and dependability of results.

Like other studies, we have some limitations in this research. First, we limited our literature search to 2016–2023 for five major chronic diseases. This study, therefore, may not include some crucial articles published before 2016 or considered conditions other than these five. Second, we considered only four classic ensemble approaches (bagging, boosting, stacking, and voting) for disease prediction. By definition, an ensemble approach combines two or more weak classifiers to generate a stronger classifier to obtain a better predictive performance than each of its constituent classifiers. Although many different ensemble approaches are in the current literature, including the ones based on time series data [66,67,68], this study emphasised the four classic techniques. A future study could fill this gap. Third, when searching current literature, this study considered Australia’s five prevalent chronic diseases. This prevalence could differ across different countries, potentially leading to a selection bias. Therefore, broader coverage of chronic diseases would make this study’s findings applicable across more countries. Fourth, there may be articles in the literature that met our selection criteria but remained unnoticed by our search criteria, potentially leading to another selection bias. The abovementioned limitations also create potential research opportunities that researchers in the future could address.

## 6. Conclusions

Various early disease prediction strategies are being developed and improved using machine learning and artificial intelligence. In machine learning, the ensemble approach is the best for disease prediction because it reduces bias and variance, improving the model’s disease prediction accuracy. Bagging, boosting, stacking, and voting are the four main ensemble approaches that are frequently used for disease prediction. According to this literature review, using an ensemble approach on other machine learning algorithms improves the accuracy of the base classifier. This paper provides a complete evaluation of the literature on disease prediction models utilising various ensemble classes and an overview of the ensemble technique. The descriptions of the literature-based publications provide essential information regarding how these algorithms fared in various combinations. We extracted the 45 papers reviewed in this study using the search approach specified in Section 3. This research can help scholars decide whether ensemble approaches are suited for their project. This research can also assist them in determining the most accurate ensemble strategy for disease prediction. We found that stacking has been used in half the studies considered for comparison with bagging and boosting, but it has shown 82.6% best performance accuracy. On the other hand, bagging has been used in most of the studies but showed the least prominent results of just 26.8%, followed by boosting, which has shown the best results 40.5% of the time.

## Figures and Tables

**Figure 1 healthcare-11-01808-f001:**
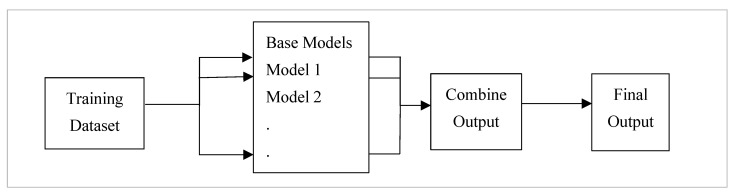
Ensemble learning in which data can be trained on different base classifiers, and the output is combined to obtain the final prediction.

**Figure 2 healthcare-11-01808-f002:**
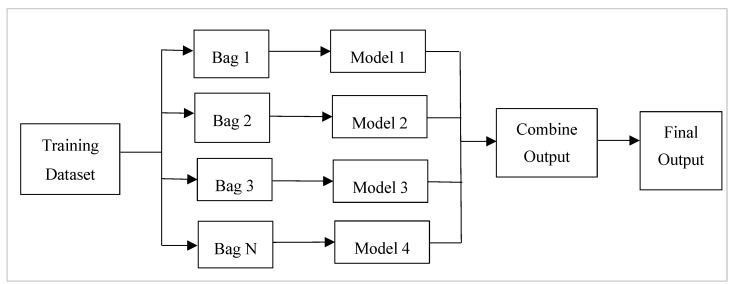
Steps followed in the bagging approach. Bags of data are formed from the input dataset, and models are used on all the bags. The output is combined from all the models.

**Figure 3 healthcare-11-01808-f003:**
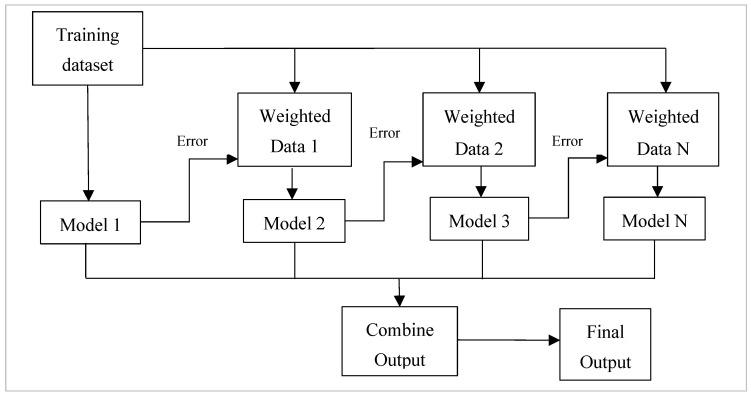
The framework used in the boosting approach. Different models are considered, and each model tries to compensate for the weakness of its predecessor by reducing the error.

**Figure 4 healthcare-11-01808-f004:**
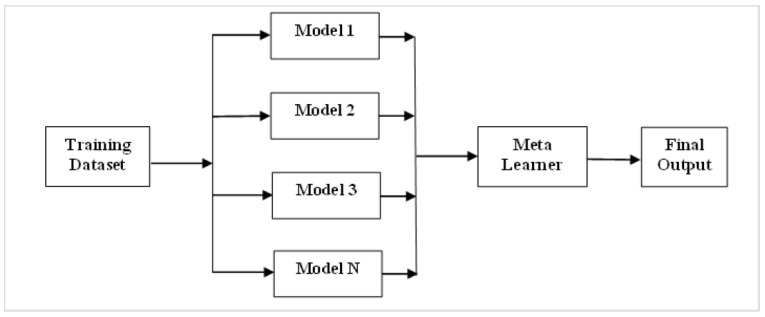
The framework for the stacking approach. Different models are used on the input dataset, and the metalearner uses the output from all the models to make the final predictions.

**Figure 5 healthcare-11-01808-f005:**
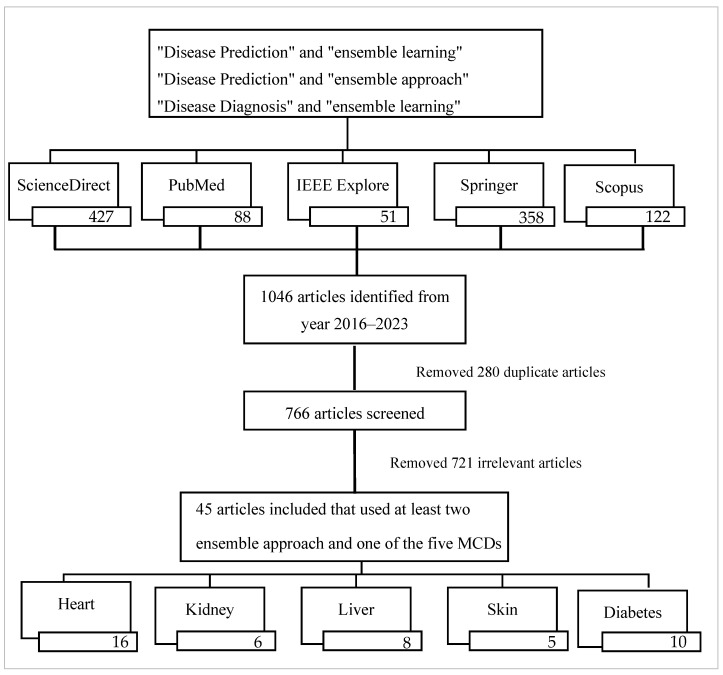
The overall data collection process and the number of articles considered for different disease categories, according to the PRISM guideline [26]. MCD, major chronic disease.

**Figure 6 healthcare-11-01808-f006:**
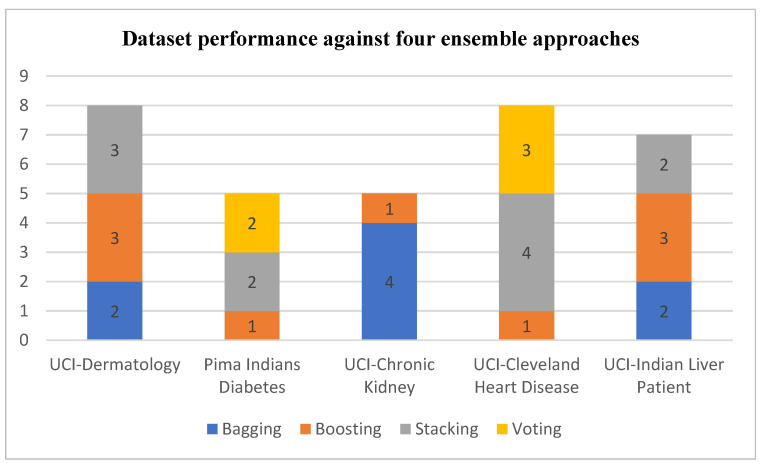
The number of times each ensemble approach has been found as the best performing against the five frequently used datasets.

**Table 1 healthcare-11-01808-t001:** Summary of reviewed techniques in Heart Disease.

Ref.	Base Learner Models	Ensemble Model	Data Type	Preprocessing Technique	Positive/Negative Cases	Dataset	Attributes/Instances	Accuracy	Best Model
[35]	KNN, LR, SVM, RF, CART, LDA	Gradient Boost, RF	Clinical		139/164	UCI Cleveland Heart Disease	14/303	Bagging (RF) = 83%, Boosting (Gradient) = 81%	Boosting
[4]	LR	RF, AdaBoost, Voting, Stacking		Random oversampling	644/3594	Kaggle Chronic Heart Disease	16/4238	Bagging (RF) = 96%, AdaBoost = 64%, Voting = 76%, Stacking = 99%	Stacking
[36]	SVM	AdaBoost, Stacking, RF	Clinical	Feature selection	139/164	UCI Cleveland Heart Disease	14/303	Bagging (RF) = 88.0%, Boosting (AdaBoost) = 88.0%, Stacking = 92.2%	Stacking
[37]	SVM	Stacking, RF	Clinical	Feature selection, Optimisation	139/164	UCI Cleveland Heart Disease	14/303	Stacking = 91.2%, Bagging (RF) = 82.9%	Stacking
[19]	XGB, LR, RF, KNN	Majority Voting, XGBoost, RF	Clinical	Feature selection	139/164	UCI Cleveland Heart Disease	14/303	Voting = 94%, Bagging (RF) = 92%, Boosting (XGBoost) = 87%	Voting
[38]	LR, SVM	RF, XGBoost	Clinical	Feature Selection,	1447/7012	Cardiovascular disease	131/8459	Bagging (RF) = 83.6%, Boosting (XGBoost) = 83.8%	Boosting
[39]	XGB, DT, KNN	Stacking, RF, XGB, DT		Eliminating outliers, Scaling		Kaggle Cardiovascular	13/7000	Stacking = 86.4%, Bagging (RF) = 88.6%, Boosting (XGBoost) = 88.1%, Bagging (DT) = 86.3%	Bagging
[40]	DT, AdaBoost, LR, SGD, RF, SVM, GBM, ETC, G-NB	DT, AdaBoost, RF, GBM	Clinical	Oversampling		UCI Heart Failure	13/299	Bagging (DT) = 87.7%, Boosting (AdaBoost) = 88.5%, Bagging (RF) = 91.8%, Boosting (GBM) = 88.5%	Bagging
[20]	LR, SVM, KNN, DT, RF	Majority Voting, RF, DT	Clinical	Handled missing values, imputation, normalisation	139/164	UCI Cleveland Heart Disease	14/303	Voting = 98.18%, Bagging (DT) = 93.1%, Bagging (RF) = 94.4%	Voting
[41]	NB, KNN, RT, SVM, BN	AdaBoost, LogitBoost, RF	Clinical			UCI SPECT heart disease	22	Bagging (RF) = 90%, Boosting (AdaBoost) = 85%, Boosting (LogitBoost) = 93%	Boosting
[7]	NB, RF, MLP, BN, C4.5, PART	Bagging, Boosting, and Stacking,	Clinical	Handled missing values	139/164	UCI Cleveland Heart Disease	14/303	Bagging = 79.87%, Boosting = 75.9% Stacking = 80.21%, Voting = 85.48%	Voting
[42]	KNN, SVM, NB, LR, QDA, C4.5, NN	Bagging, AdaBoost, and Stacking	Clinical		139/164	UCI Cleveland Heart Disease	14/303	Bagging = 77.9%, Boosting (AdaBoost) = 64.3%, Stacking = 82.5%	Stacking
[5]	LR, KNN, SVM, DT, NB, MLP	Bagging, Boosting, and Stacking			Equal	Kaggle Cardiovascular Disease	12/-	Bagging = 74.42%, Boosting = 73.4%, Stacking = 75.1%	Stacking
[43]	RF, ET, XGBoost, GB	AdaBoost, GBM, Stacking		Eliminated outliers		IEEE Data Port	11/1190	Boosting (GBM) = 84.2%, Boosting (AdaBoost) = 83.4%, Stacking = 92.3%	Stacking
[44]	MLP, SCRL, SVM	Bagging, Boosting, and Stacking	Clinical		139/164	UCI Cleveland Heart Disease	14/303	Bagging = 80.5%, Boosting = 81.1%, Stacking = 84.1%	Stacking
[28]	DT, CNN, NB, ANN, SVM, CAFL	Bagging, Boosting		Data distribution		Eric	7/210	Bagging = 73.2%, Boosting (AdaBoost) = 65.1%, Stacking = 79.4%	Stacking

**Table 2 healthcare-11-01808-t002:** Summary of reviewed techniques in Chronic Kidney Disease.

Ref.	Base Learner Models	Ensemble Model	Data Type	Preprocessing Technique	Positive/Negative Cases	Dataset	Attributes/Instances	Accuracy	Best Model
[45]	NB, LR, MLP, SVM, DS, RT	AdaBoost, Bagging, Voting, Stacking	Clinical	Handled missing values, feature selection, and sample filtering		Razi Hospital	42/936	Bagging = 99.1%, Boosting (AdaBoost) = 99.1%, Voting = 96.6%, Stacking = 97.1%	Boosting Bagging
[46]	NB, LR, ANN, CART, SVM	Gradient Boosting, RF	Clinical	Feature selection, handling missing values, and imputation	250/150	UCI Chronic Kidney	25/400	Bagging (RF) = 96.5%, Boosting (Gradient Boosting) = 90.4%	Bagging
[2]	-	AdaBoost, RF, ETC bagging, Gradient boosting	Clinical	Feature engineering	250/150	UCI Chronic Kidney	25/400	Bagging (Extra trees) = 98%, Bagging = 96%, Bagging (RF) = 95%, Boosting (AdaBoost) = 99%, Boosting (Gradient) = 97%	Boosting
[47]	LR, KNN, SVC	Gradient Boosting, RF	Clinical	Handled missing values	250/150	UCI Chronic Kidney	25/400	Bagging (RF) = 99%, Boosting (Gradient) = 98.7%	Bagging
[13]	-	AdaBoost, Bagging and Random Subspaces	Clinical	Feature extraction	250/150	UCI Chronic Kidney	25/400	AdaBoost = 99.25%, Bagging = 98.5%, Bagging (Random Subspace) = 99.5%	Bagging
[11]	NB, SMO, J48, RF	Bagging, AdaBoost		Feature selection and handling missing values	250/150	UCI Chronic Kidney	25/400	Bagging = 98%, Bagging (RF) = 100%, Boosting (AdaBoost) = 99%	Bagging

**Table 3 healthcare-11-01808-t003:** Summary of reviewed techniques in Skin Cancer. The positive/negative cases come in a range since this dataset contains data for six different skin cancer conditions.

Ref.	Base Learner Algorithm	Ensemble Approach	Data Type	Preprocessing Technique	Positive/Negative Cases	Dataset	Attributes/Instances	Accuracy	Best One
[18]	NB, RF KNN, SVM, and MLP	Bagging, Boosting, and Stacking	Clinical	Handled missing values	[20–112]/[254–346]	UCI Dermatology	34/366	Bagging = 96%, Boosting = 97%, Stacking = 100%	Stacking
[8]	DT, LR	Bagging, AdaBoost, and Stacking	Clinical	Feature selection	[20–112]/[254–346]	UCI Dermatology	34/366	Bagging = 92.8%, Boosting (AdaBoost) = 92.8%, Stacking = 92.8%	Bagging Boosting Stacking
[48]	LR, CHAID DT	Bagging, Boosting	Clinical	Handled missing values, data distribution, and balancing,	[20–112]/[254–346]	UCI Dermatology	34/366	Bagging = 100%, Boosting = 100%	Bagging Boosting
[31]	NB, KNN, DT, SVM, RF, MLP	Bagging, Boosting, and Stacking	Clinical	Hybrid Feature selection, information gain, and PCA	[20–112]/[254–346]	UCI Dermatology	12/366	Bagging = 95.94%, Boosting = 97.70%, Stacking = 99.67%	Stacking
[16]	PAC, LDA, RNC, BNB, NB, ETC	Bagging, AdaBoost, Gradient Boosting	Clinical	Feature Selection	[20–112]/[254–346]	UCI Dermatology	34/366	Bagging = 97.35%, AdaBoost = 98.21%, Gradient Boosting = 99.46%	Boosting

**Table 4 healthcare-11-01808-t004:** Summary of reviewed techniques in Liver Disease.

Ref.	Base Learner Models	Ensemble Model	Data Type	Preprocessing Technique	Positive/Negative Cases	Dataset	Attributes/Instances	Accuracy	Best Model
[3]	BeggRep, BeggJ48, AdaBoost, LogitBoost, RF	Bagging, Boosting	Clinical		416/167	UCI Indian Liver Patient	10/583	Boosting(AdaBoost) = 70.2%, Boosting(LogitBoost) = 70.53%, Bagging (RF) = 69.2%	Boosting
[49]	NB, SVM, KNN, LR, DT, MLP	Stacking, DT	Clinical	Feature Selection PCA	416/167	UCI Indian Liver Patient	10/583	Bagging (DT) = 69.40% Stacking = 71.18%	Stacking
[50]	KNN	RF, Gradient Boosting, AdaBoost, Stacking	Clinical		416/167	UCI Indian Liver Patient	10/583	Bagging (RF) = 96.5%, Boosting(Gradient) = 91%, Boosting(AdaBoost) = 94%, Stacking = 97%	Stacking
[33]	DT, NB, KNN, LR, SVM, AdaBoost, CatBoost	XGBoost, Light GBM, RF	Clinical	Handled missing values	416/167	UCI Indian Liver Patient	10/583	Bagging (RF) = 88.5% Boosting(XGBoost) = 86.7% Boosting (LightGBM) = 84.3%	Bagging
[32]	SVM, KNN, NN, LR, CART, ANN, PCA, LDA	Bagging, Stacking	Clinical	Handled missing values, feature selection, PCA	453/426	Iris And Physiological	22/879	Bagging (RF) = 85%, Stacking = 98%	Stacking
[9]	KNN, SVM, RF, LR, CNN	RF, XGBoost, Gradient Boost		Handled missing values, scaling, and feature selection		Image	11/10,000	Bagging (RF) = 83% Boosting (XGBoost) = 82% Boosting (Gradient) = 85%	Boosting
[51]	LR, DT, RF KNN, MLP	AdaBoost, XGBoost, Stacking	Clinical	Data Imputation, label encoding, resampling, eliminating duplicate values and outliers	416/167	UCI Indian Liver Patient	10/583	Boosting (AdaBoost) = 83% Boosting (XGBoost) = 86% Stacking = 85%	Boosting
[10]	DT, KNN, SVM, NB	Bagging, Boosting, RF	Clinical	Discretisation, resampling, PCA	416/167	UCI Indian Liver Patient	10/583	Bagging (RF) = 88.6%, Bagging = 89%, Boosting = 89%	Bagging Boosting

**Table 5 healthcare-11-01808-t005:** Summary of reviewed techniques in diabetes.

Ref.	Base Learner Models	Ensemble Model	Data Type	Preprocessing Technique	Positive/Negative Cases	Dataset	Attributes/Instances	Accuracy	Best Model
[52]	MLP, SVM, DT, LR	RF, Stacking		Outlier detection and elimination, SMOTE Tomek for imbalanced data	73/330	Type 2 Diabetes	19/403	Bagging (RF) = 92.5% Stacking = 96.7%	Stacking
[6]	RF, LR, NB	Soft voting classifier, AdaBoost, Bagging, XGBoost,	Clinical	Min–max normalisation, label encoding, handled missing values	268/500	Pima Indians Diabetes	9/768	Bagging = 74.8%, Boosting(AdaBoost) = 75.3%, Boosting (XGBoost) = 75.7%, Voting = 79.0%	Voting
[53]	SVM, KNN, DT	Bagging, Stacking	Clinical	SMOTE, k-fold cross validation		KFUH Diabetes	10/897	Bagging = 94.1%, Stacking = 94.4%	Stacking
[54]	KNN, LR, MLP	AdaBoost, Stacking		Feature selection, handling missing values	60/330	Vanderbilt University’s Biostatistics program	18/390	Boosting (AdaBoost) = 91.3%, Stacking = 93.2%	Stacking
[55]	XGB, CGB, SVM, RF, LR	XGBoost, RF, CatBoost		Missing values eliminated, class imbalance handling, feature selection	33,332/73,656	NHANES	18/124,821	Boosting(XGBoost) = 70.8%, Bagging(RF) = 78.4%, Boosting(CatBoost) = 82.1%	Boosting
[38]	LR, SVM	RF, XGBoost	Clinical	feature Selection	5532/15,599	Cardiovascular disease	123/21,131	Bagging (RF) = 85.5%, Boosting (XGBoost) = 86.2%	Boosting
[17]	SVM	Majority voting, stacking	Clinical	Cross-validation	268/500	Pima Indians Diabetes	9/768	Stacking = 79%, Voting = 65.10%	Stacking
[22]	ANN, SVM, KNN, NB	Bagging, RF, Majority Voting	Clinical		268/500	Pima Indians Diabetes	9/768	Bagging(RF) = 90.97%, Bagging = 89.69%, Voting = 98.60%,	Voting
[34]	KNN, RF, DT, SVM, MLP, GB	RF, AdaBoost, Stacking	Clinical	Feature selection with genetic algorithm	268/500	Pima Indians Diabetes	9/768	Bagging (RF) = 93%, Boosting (GBC) = 95%, Stacking = 98.8%	Stacking
[10]	DT, KNN, SVM	Bagging, Boosting, RF	Clinical	Discretisation, resampling, PCA	268/500	Pima Indians Diabetes	9/768	Bagging (RF) = 89.7%, Bagging = 89.5%, Boosting = 90.1%	Boosting

**Table 6 healthcare-11-01808-t006:** Advantages and limitations of the four ensemble classes considered in this study.

Ensemble Approach	Advantages	Limitations
Bagging	- One benefit of bagging is that it enables several weak learners to work together to outperform a single strong student. - The variance in performance measures is greatly reduced by bagging while keeping the bias almost the same [3]. - Since sampling is performed by bootstrapping, the training data become more diversified, making this approach effective [10]. - If the training set is particularly large, training the model on a smaller data set can still improve model accuracy while saving computation time.	- The fundamental drawback of bagging is that it increases model accuracy at the expense of interpretability [8]. For example, if a single tree were used as the base model, the diagram would be more appealing and easier to understand, but when bagging is employed, this interpretability is lost. - Another drawback is that we cannot determine which features are being chosen during sampling, which increases the likelihood that crucial data may be lost if some characteristics are never used [10].
Boosting	- As an ensemble model, boosting has a simple-to-read and easy-to-comprehend algorithm, making it simple to interpret its predictions [5]. - Boosting is a robust technique that efficiently reduces overfitting [3].	- Boosting has the drawback of being sensitive to outliers because every classifier is required to correct the mistakes made by the predecessors. As a result, the technique is overly reliant on outliers [3]. - It is challenging to streamline the process because every estimator rests its accuracy on prior predictions.
Stacking	- The advantage of stacking is that it can use a variety of effective models to accomplish classification or regression tasks and produce predictions that perform better than any one model in the ensemble. - Stacking increases the precision of model prediction.	- Huge datasets will require more computation time since each classifier must individually process the entire dataset during training, which increases computational time [5].
Voting	- Voting takes advantage of distinct classifiers’ strengths while compensating for their flaws, resulting in increased performance [56]. - The voting can handle various data types, including categorical, numerical, and text data, allowing for a more in-depth examination [57]. - It can also provide insights into decision-making by analysing voting patterns [21].	- More complex, sensitive to correlated classifiers, lack of interpretability, and the potential for overfitting [19]. - The individual classifiers may not significantly improve accuracy if they are closely connected. - Classifier diversity is critical for improved performance. - It may overfit training data if they are not correctly regularised [7].

**Table 7 healthcare-11-01808-t007:** Comparison of usage frequency and best performance of ensemble classes considered in this study.

Ensemble Approach	Number of Published Articles Used This Algorithm	Number of Times This Algorithm Showed the Best Performance (%)
Bagging	41	11 (26.8%)
Boosting	37	15 (40.5%)
Stacking	23	19 (82.6%)
Voting	7	5 (71.4%)

**Table 8 healthcare-11-01808-t008:** Comparison of usage frequency and best performance of ensemble classes against the five major chronic diseases considered in this study.

Disease Name	Total Reviewed Article	Bagging		Boosting		Stacking		Voting	
		Usage Frequency	Best Performance	Usage Frequency	Best Performance	Usage Frequency	Best Performance	Usage Frequency	Best Performance
Heart disease	16	15	2 (13.4%)	14	3 (21.4%)	10	8 (80%)	4	3 (75%)
Kidney disease	6	6	5 (83.4%)	6	2 (33.4%)	1	0 (0%)	0	0 (0%)
Skin Cancer	5	5	2 (40%)	5	3 (60%)	3	3 (100%)	0	0 (0%)
Liver disease	8	7	2 (28.5%)	6	4 (66.7%)	4	3 (75%)	0	0 (0%)
Diabetes disease	10	8	0 (0%)	6	3 (50%)	5	5 (100%)	3	2 (66.7%)
Total	45	41	11 (26.8%)	37	15 (40.5%)	23	19 (82.6%)	7	5 (71.4%)

## Data Availability

Data can be shared upon request.

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
