# Peer review of "Ensemble Learning for Disease Prediction: A Review"

_healthcare, 2023, doi:10.3390/healthcare11121808_

Round 1
Reviewer 1 Report (New Reviewer)
This review scrutinizes the prevailing literature on the application of ensemble machine learning methods - bagging, boosting, and stacking - for disease prediction. As an increasingly prominent area of research, these ensemble methods have demonstrated significant potential.
Addressing the below-mentioned comments would strengthen the overall quality and comprehensiveness of your review paper.
#1. The authors could provide a more in-depth analysis of each method (bagging, boosting, and stacking) and their applications in different disease contexts. For example, it would be helpful to understand why stacking outperforms bagging and boosting in certain disease prediction tasks. What are the specific characteristics of these diseases or the datasets that make stacking more suitable?
#2. The authors have focused on three ensemble methods - bagging, boosting, and stacking - which are indeed popular and widely used. However, there are other ensemble methods, such as random forests and voting classifiers, that also show promise in disease prediction. It would be valuable to see these methods included in the review, providing a more holistic view of the field.
#3. The authors have identified 42 studies for this review. While this is a good number, the authors should evaluate and discuss potential selection bias in their review process. Were there any specific criteria used for study selection that may have led to the overrepresentation of studies favouring a particular ensemble method? Additionally, the authors should also consider discussing the quality of the studies included in the review.
#4. While the focus on prediction accuracy is understandable and essential, the interpretability of the models and their applicability in a clinical setting is equally important. The authors should discuss these aspects for each of the ensemble methods reviewed. For instance, while stacking might have shown better performance, it might also be more complex and less interpretable than simpler methods like bagging. This trade-off between performance and interpretability should be discussed in the review.
#5. Data preprocessing is a crucial step in machine learning, and it may significantly impact the performance of the ensemble methods. It would be beneficial if the authors could provide more information on the preprocessing steps taken in the studies reviewed. Were there any common preprocessing steps or challenges across the studies? How might these preprocessing steps have influenced the results?
I recommend thorough proofreading and language editing to ensure the paper adheres to standard English conventions.
Author Response
Comment 1
The authors could provide a more in-depth analysis of each method (bagging, boosting, and stacking) and their applications in different disease contexts. For example, it would be helpful to understand why stacking outperforms bagging and boosting in certain disease prediction tasks. What are the specific characteristics of these diseases or the datasets that make stacking more suitable?
Our response
Thank you for highlighting this point. Stacking is a popular method for combining predictions from base models, uncovering hidden patterns, and capturing nonlinear interactions in heart, skin, liver, and diabetic illnesses. It takes advantage of each model's strengths while limiting data heterogeneity. The significance of stacking is determined by the dataset, clinical variables, and modelling choices. For further details, please see lines 386-397 on page 18. We did not notice any specific dataset characters impacting ensemble performance. We considered dataset characteristics of data size and attribute, positive and negative instances, pre-processing techniques followed and data type (Tables 1-5) but did not find any significant impact on ensemble performance.
Comment 2
The authors have focused on three ensemble methods - bagging, boosting, and stacking - which are indeed popular and widely used. However, there are other ensemble methods, such as random forests and voting classifiers, that also show promise in disease prediction. It would be valuable to see these methods included in the review, providing a more holistic view of the field.
Our response
We have considered the voting approach in this revised manuscript. We describe this approach in lines 215-246 on pages 8-9 in the revised manuscript. Due to this inclusion (of the voting approach), the numerical values of all our previous findings have been changed. Please see the different tables and figures for this change. The last row of Table 6 on page 17 highlights the advantages and limitations of the voting approach. We considered random forest as a bagging approach. For this reason, we did not consider it as a separate one.
Comment 3
The authors have identified 42 studies for this review. While this is a good number, the authors should evaluate and discuss potential selection bias in their review process. Were there any specific criteria used for study selection that may have led to the overrepresentation of studies favouring a particular ensemble method? Additionally, the authors should also consider discussing the quality of the studies included in the review.
Our response
We made a revised search based on the comment by another reviewer. We now reviewed 45 articles in this revised submission. Please see Tables 1 and 5 (highlighted rows) for the newly added three reviewed articles. Factors, such as overrepresentation and selection bias, could impact this literature review. However, these factors could only impact the numerical values of our findings and cannot impact their orders. We discuss their potential impact in lines 467-481 on page 21.
Comment 4
While the focus on prediction accuracy is understandable and essential, the interpretability of the models and their applicability in a clinical setting is equally important. The authors should discuss these aspects for each of the ensemble methods reviewed. For instance, while stacking might have shown better performance, it might also be more complex and less interpretable than simpler methods like bagging. This trade-off between performance and interpretability should be discussed in the review.
Our response
Thank you for highlighting this point which will provide more useful information. The applicability and interpretability of each technique have been added to the revised manuscript. For further details, please see lines 398-405 on pages 18-19.
Comment 5
Data preprocessing is a crucial step in machine learning, and it may significantly impact the performance of the ensemble methods. It would be beneficial if the authors could provide more information on the preprocessing steps taken in the studies reviewed. Were there any common preprocessing steps or challenges across the studies? How might these preprocessing steps have influenced the results?
Our response
Thank you for asking this question. We have included a new column for pre-processing techniques used in the articles in the summary section. Please see Tables 1-5 for details. We further discuss the implication of different data preprocessing approaches in lines 437-445 on page 20.
Reviewer 2 Report (New Reviewer)
I would appreciate the authors for the prepared paper. Generally, the idea of the paper is nice and could be sufficient for publication in this journal; however, I think the authors can improve the paper by the following suggestions.
Major comments:
1- The paper does not include sufficient information about the datasets. Please provide one section about the related dataset. Discuss their features, number of samples and other related fields. Also, discuss whether they are open access or not. It should be clarified how the data were gathered in these studies. Also, you can attach the open access data in the supplementary materials.
2- I find some papers that had not been cited in the paper. Can you clarify the reason?
https://www.ncbi.nlm.nih.gov/pmc/articles/PMC7931939/
https://journals.sagepub.com/doi/full/10.1177/20552076231173225
Minor comments:
1- Remove the typos in Table 4.
2- Discuss about the application of ensemble learning in timeseries and cite the following papers.
https://onlinelibrary.wiley.com/doi/full/10.1002/eng2.12486
https://www.sciencedirect.com/science/article/abs/pii/S0378779621005654
https://www.sciencedirect.com/science/article/abs/pii/S1568494621005251?via%3Dihub
https://www.hindawi.com/journals/cin/2016/6212684/
NA
Author Response
Comment 1.
The paper does not include sufficient information about the datasets. Please provide one section about the related dataset. Discuss their features, number of samples and other related fields. Also, discuss whether they are open access or not. It should be clarified how the data were gathered in these studies. Also, you can attach the open-access data in the supplementary materials.
Our response
Thank you for the suggestion. In the revised manuscript, we have added five new features of the datasets used in our reviewed articles. Please see columns 4-8 of Tables 1-5 for further information. We further discuss the performance of different ensemble approaches against the frequently used five datasets. We noticed some interesting observations summarised in Figure 6 on page 19 and lines 415-425 on pages 19-20.
Comment 2
I found some papers that had not been cited in the paper. Can you clarify the reason?
https://www.ncbi.nlm.nih.gov/pmc/articles/PMC7931939/
https://journals.sagepub.com/doi/full/10.1177/20552076231173225
Our response
The search strategy in this study was designed to find articles that had used at least two of the three ensemble approaches (bagging, boosting, stacking, and voting) and had looked at the five main chronic diseases being studied: heart disease, liver disease, skin diseases, diabetes, and kidney disease. These two papers were excluded from consideration because the analyses were based on a medical dataset and eventually did not satisfy our predetermined search criteria. Our study focuses on chronic diseases and the ensemble method-based prediction of those diseases, so studies that used datasets unrelated to the targeted diseases were excluded. The search strategy ensured valuable insights tailored to the illnesses and ensemble methods under investigation.
Comment 3
Remove the typos in Table 4.
Our response
Thank you for mentioning this. Typos have been removed from that Table which is now Table 7 in the revised version.
Comment 4
Discuss the application of ensemble learning in time series and cite the following papers.
https://onlinelibrary.wiley.com/doi/full/10.1002/eng2.12486
https://www.sciencedirect.com/science/article/abs/pii/S0378779621005654
https://www.sciencedirect.com/science/article/abs/pii/S1568494621005251?via%3Dihub
https://www.hindawi.com/journals/cin/2016/6212684/
Our response
We discuss the implication of these articles to our study context and cite them accordingly. Please see lines 472-474 on page 21 for more details.
Reviewer 3 Report (New Reviewer)
The study investigates the performance of three ensemble techniques (bagging, boosting, and stacking) in predicting five diseases (diabetes, skin disease, kidney disease, liver disease, and heart conditions). A thorough literature search identified 42 relevant articles published between 2016 and 2022. However, the following are my comments:
1. In abstract, the sentence: “.. applied two or more of 18 the three ensemble approaches to any of these five diseases and were published in 2016-22”. Should write 22 as year format (i.e. 2022)”. Why only 2022? My suggestion it is better to cover the studies published in 2023 as well to be up-to-date.
2. The authors should clearly state what are the problems, objectives, contributions and the novel aspects of the paper.
3. In figure 5, articles after duplicates removed 42. I think many inclusion and exclusion criteria Processes (steps) should be stated clearly. I belief it is not only duplicates removed, authors should explain the steps were done exactly.
4. In the same figures authors should follow and apply PRISM for reporting in systematic reviews and meta-analyses. The current figure seems like PRISM but not proper.
5. Inclusion and exclusion criteria should be shown clearly in table(s).
6. The authors should add some figures based on the results obtained.
7. Please elaborate more analysis with latest references and should show provide critical issues in evaluation techniques.
8. I suggest authors to add new section about the analysis of the selected studies and show the challenges and open research issues.
9. There are reputation of the disease names in table 1, authors should remove the Third columns since the name of the disease name already in the table title.
10.Authors should show the details of how is the current ensemble techniques improve the performance of the mentioned five diseases.
11.To provide valuable information to the readers, authors should show the dataset
12.Based on the output of the study, please explain clearly what the exact significance of the ensemble methods to the medical diagnosis.
Thank you for the work.
Some minor editing of English language required.
Author Response
Comment 1
In abstract, the sentence: “.. applied two or more of 18 the three ensemble approaches to any of these five diseases and were published in 2016-22”. Should write 22 as year format (i.e. 2022)”. Why only 2022? My suggestion is that it is better to cover the studies published in 2023 as well to be up-to-date.
Our response
Thank you for giving the suggestion. Firstly, the abstract’s typo “2016-22” is corrected. Also, we have included studies published in 2023 to keep this study up-to-date. We queried using our search phrases on 30-05-2023 (lines 268-269). Due to this new timeline, we ended up with 45 articles. Previously, this figure was 42. This new inclusion numerically impacts all our previous findings. Please see the different highlighted texts of the manuscript for further details.
Comment 2
The authors should clearly state what are the problems, objectives, contributions and the novel aspects of the paper.
Our response
Thank you for this comment. We revised the abstract to state most of our study's above-mentioned purposes. On top of this, we conducted further analyses to extract novel aspects of our findings (e.g., Figure 6 on page 19 and its corresponding texts in lines 415-425). We further discussed various aspects of our study and its relevant findings in the discussion section. For example, we stressed on how data preprocessing impact ensemble performance (lines 437-445) and the tradeoff between ensemble complexity and interpretation of findings (lines 386-405).
Comment 3
In Figure 5, articles after duplicates were removed 42. I think many inclusion and exclusion criteria Processes (steps) should be stated clearly. I believe it is not only duplicates removed, authors should explain the steps that were done exactly.
Our response
Thank you for this comment. We added further info in Figure 5 to clarify our search approach. We further added textual descriptions for our literature search. For further details, please see lines 277-281 on pages 9-10.
Comment 4
In the same figure, authors should follow and apply PRISM for reporting in systematic reviews and meta-analyses. The current figure seems like PRISM but not proper.
Our response
Figure 5 has been updated, mentioning the selected articles' clear inclusion and exclusion criteria. We also mentioned this in the caption.
Comment 5
Inclusion and exclusion criteria should be shown clearly in table(s).
Our response
They have been added in Figure 5 and lines 277-281 in the revised version of the manuscript.
Comment 6
The authors should add some figures based on the results obtained.
Our response
We added more figures (e.g., Figure 6 on page 19) as well as elaborated our search to improve the overall quality of our manuscript. We also added more articles to review to make the review more comprehensive and up-to-date. Please see the highlighted rows of Tables 1 and 5 for further details.
Comment 7
Please elaborate more analysis with the latest references and should show provide critical issues in evaluation techniques.
Our response
We conducted more analyses and added more recent references. Please see Tables 1 and 5 for more recent references. Please see the results and discussion sections for further analyses and their related discussion.
Comment 8
I suggest authors add a new section about the analysis of the selected studies and show the challenges and open research issues.
Our response
We now organised the results section and added new sections related to analysing selected articles. For example, we added Section 4.3 (Ensemble performance against datasets) on page 19 to demonstrate how our four ensemble approaches perform against frequently used datasets.
Comment 9
There are repetitions of the disease names in Table 1, authors should remove the Third column since the name of the disease name already in the table title.
Our response
Thank you for noticing this. We have removed this column from Tables 1-5.
Comment 10
The authors should show the details of how the current ensemble techniques improve the performance of the mentioned five diseases.
Our response
The study looks at how ensemble strategies (bagging, boosting, stacking, and voting) function in five diseases: diabetes, skin disease, kidney illness, liver disease, and heart disorders. In evaluated research, stacking is the best ensemble strategy for diabetes prediction, obtaining 100% accuracy. Stacking is the process of combining predictions from numerous base models to capture hidden patterns and nonlinear relationships. Voting has also been used to predict diabetes, with a 66.7% accuracy rate. Stacking is helpful for predicting skin diseases and obtaining 100% accuracy. Stacking and voting were not used in the prediction of kidney illness, although bagging approach yielded accuracy rate of 83.4%. Stacking is useful for predicting liver illness because it captures diverse predictions from base models while uncovering hidden patterns and intricate linkages. Stacking has demonstrated encouraging results in the prediction of cardiac disease. Stacking has showed encouraging results in the prediction of heart disease, with the best accuracy rate of 75%. Bagging has not performed well in the prediction of heart disease, obtaining the best performance in 13.4% of cases. Although voting has been used in fewer trials, it has shown good accuracy rates for predicting heart and diabetes illness.
Comment 11
To provide valuable information to the readers, authors should show the dataset.
Our response
Thank you for this suggestion. We added columns for dataset information in Tables 1-5.
Comment 12
Based on the output of the study, please explain clearly the exact significance of the ensemble methods to the medical diagnosis.
Our response
Bagging, boosting, stacking, and voting are important medical diagnosis and disease prediction approaches. They boost accuracy by merging many classifiers, capturing various patterns and information, minimising bias and variation, and discovering hidden patterns and linkages. These strategies enable machine learning models to overcome bias and variance concerns, resulting in more robust and trustworthy illness prediction models. Furthermore, ensemble approaches can find intricate and nonlinear correlations between risk factors, symptoms, and diseases, making them a valuable tool in medical diagnosis. Ensemble approaches improve disease detection and prediction by tackling data heterogeneity and class imbalance difficulties in medical datasets. They address changes in data attributes and uneven class distributions by merging models trained on multiple subsets, ensuring accurate disease prediction across diverse subgroups and minority cases. Clinical applicability is also critical, with ensemble approaches providing a good blend of performance and interpretability. These technologies allow healthcare providers to make more accurate diagnoses, identify high-risk individuals, and eventually improve individual health outcomes and care costs. We have mentioned texts like these and others in the introduction and discussion sections of the main manuscript. Please see the highlighted texts in these two sections.
Reviewer 4 Report (New Reviewer)
The authors came up with a review paper summarizing ensemble technique applied for disease prediction. Following are few suggestions to improve the paper:
1. The authors define ensemble methods as the best way to identify disease (line 340), but the paper does not throw light on practices that may hinder its use. Data leakage is possible through ensemble methods, are there any ways being used to tackle it?
2. They have summarized various articles classified at disease level, specified the type of ensemble technique used. If it is possible, they should also provide summary of recent datasets for each type of disease classification. This would help understand the state of art for individual machine learning models.
3. In all, the paper lacks the ability to address the issue from the aspect of machine learning. The paper would act as a source of reference, if they are able to provide an example of machine learning pipeline, and how is that implemented for disease prediction.
Author Response
Comment 1
The authors define ensemble methods as the best way to identify the disease (line 340), but the paper does not throw light on practices that may hinder its use. Data leakage is possible through ensemble methods, are there any ways being used to tackle it?
Our response
Thank You for raising this point, researchers have used cross-validation, train-validation-test split, feature engineering precautions, and model stacking precautions to solve data leaking concerns in ensemble approaches for disease detection. These security measures ensured the precise separation of training and testing data, lowering the danger of data leakage and maintaining the legitimacy and reliability of outcomes. We have included a summary in the discussion section on how data leakage has been tackled in the studies. We address the data leakage issue that can impact ensemble performance. We discussed this accordingly – please see lines 458-466 on page 21. On the same page, other possible factors have been discussed in lines 467-481.
Comment 2
They have summarized various articles classified at the disease level and specified the type of ensemble technique used. If possible, they should also provide a summary of recent datasets for each type of disease classification. This would help understand the state-of-the-art for individual machine learning models.
Our response
Instead of having separate tables, we added comprehensive attribute descriptions for datasets used in our reviewed articles to the existing tables. We added five columns for dataset descriptions in Tables 1-5. These columns provided further details of each dataset used in our reviewed articles. We further discuss the performance of different ensemble approaches against the frequently used five datasets. We noticed some interesting observations summarised in Figure 6 on page 19 and lines 415-425 on pages 19-20.
Comment 3
In all, the paper lacks the ability to address the issue from the aspect of machine learning. The paper would act as a source of reference if they are able to provide an example of a machine learning pipeline, and how is that implemented for disease prediction.
Our response
Our research primarily aims to provide a broad picture of ensemble approaches of bagging, boosting, stacking, and voting for predictive disease analytics. In doing so, we focus on five major chronic diseases and the predictive performance of ensemble methods on them. We discuss dataset characteristics and explore whether these characteristics impact disease predictions. This manuscript will serve as a reference source for selecting ensemble methods for disease analytics for the design of any future relevant studies. We explore which ensemble is more appropriate for which diseases.
Round 2
Reviewer 3 Report (New Reviewer)
- I would like to appreciate the efforts made by the authors to improve the quality of the paper. There are still minor comments from my previous comment report.
- Authors answer comment (5) by adding the Inclusion and exclusion criteria to the PRISM figure. My comments about to add the Inclusion and exclusion criteria should be shown clearly in table(s).
- Authors should add all their response to the comments to the paper, I can’t see the comments 10 and 12 in the revised version. It looks like the authors just answer my comments, therefore, should be reelected to the revised version in details (please make sure all the response to the reviewers should be in the revised paper).
- I urge authors to use visualize the results.
Thank you for the work.
- I would like to appreciate the efforts made by the authors to improve the quality of the paper. There are still minor comments from my previous comment report.
- Authors answer comment (5) by adding the Inclusion and exclusion criteria to the PRISM figure. My comments about to add the Inclusion and exclusion criteria should be shown clearly in table(s).
- Authors should add all their response to the comments to the paper, I can’t see the comments 10 and 12 in the revised version. It looks like the authors just answer my comments, therefore, should be reelected to the revised version in details (please make sure all the response to the reviewers should be in the revised paper).
- I urge authors to use visualize the results.
Thank you for the work.
Author Response
Comment 1
The authors answer comment (5) by adding the Inclusion and exclusion criteria to the PRISM figure. My comments about adding the Inclusion and exclusion criteria should be shown clearly in table(s).
Our response
The inclusion and exclusion criteria have already been presented in Figure 5 in full detail. In addition, the inclusion criteria are shown in Tables 1-5. The inclusion criteria are presented in the heading, mentioning that the articles must have analysed five major chronic diseases considered in the study. Also, another inclusion criterion is that it must have the analyses using at least two ensemble approaches. Column 3 of Tables 1-5 shows that the articles have used two or more approaches considered in those studies.
Comment 2
Authors should add all their responses to the comments to the paper, I can’t see comments 10 and 12 in the revised version. It looks like the authors just answered my comments, therefore, should be reelected to the revised version in detail (please make sure all the responses to the reviewers should be in the revised paper).
Our response
Please see lines 362-413 on pages 17-19 (Subsection 4.2 Frequency and accuracy comparison) for the response to comment 10 of the last revision round. For comment 12, please see lines 446-457 on pages 20-21 for our response.
Comment 3
I urge authors to use visualize the results.
Our response
We used both tables and figures to present our findings. However, we did not use both to present the same findings. For example, we used Figure 8 (line 410 on page 19) to illustrate dataset performance against four ensemble approaches. We can do the same using a Table, which would be redundant. Similarly, we used Table 8 to compare the usage frequency of ensemble approaches. We can replace this table using a figure. However, using a table to present this finding would be a better approach.
Reviewer 4 Report (New Reviewer)
The authors have done a commendable work to address the questions posed by the reviewers. There is still a minor improvement recommended: at many places the references are not cited.
For example: Section 4.3, lines 444-446, all the dataset sources can be cited.
Similarly, for data Leakage discussion, they could cite papers which are using a particular type of strategy.
Author Response
Comment 1
There is still a minor improvement recommended: in many places, the references are not cited. For example Section 4.3, lines 444-446, all the dataset sources can be cited.
Our response
Thank you for this suggestion. We cited them in this revised submission. Please see lines 416-419 on page 19 for more details.
Comment 2
Similarly, for data Leakage discussion, they could cite papers that are using a particular type of strategy.
Our response
We also added relevant citations. Please see lines 459-461 on page 21 for details. In addition, we checked the entire manuscript for any missing references.
This manuscript is a resubmission of an earlier submission. The following is a list of the peer review reports and author responses from that submission.
Round 1
Reviewer 1 Report
The quality of the paper has improved. Thanks for addressing my comments.
Author Response
Thank you for your positive comment.
Reviewer 2 Report
Dear authors,
Thank you very much for your responses.
After revising them, it is my belief that not all of my comments and concerns were properly addressed, with the detail they should have been.
Most of the responses the Authors provide are vague and imprecise, or they comment facts that are evident.
Therefore, I invite the Authors to go over all of my questions again, and elaborate meaningful answers to them.
I also recommend the Authors to help the Reviewers, by adding in the response letter the line numbers of those lines where the concerned changes were made.
As a conclusion, I encourage the Authors to continue the process for improving the manuscript, by following the indications provided and properly answering the questions planted in my former reviews.
